# Practice variation in induction of labor: A critical document analysis on the contribution of regional protocols

**Dirkje C. Zondag**[1]*, **Pien M. Offerhaus**[2], **Judit K. J. Keulen**[2], **Tamar M. van Haaren–ten Haken**[2], **Marianne J. Nieuwenhuijze**[1,2]

1 CAPHRI, Maastricht University, Maastricht, the Netherlands, 2 Research Centre for Midwifery Science, Zuyd University, Maastricht, the Netherlands

* lianne_zondag@hotmail.com

## Abstract

### Rationale

Despite national guidelines with recommendations on induction of labor (IOL), large variation in the use of this intervention exists between regions in the Netherlands. Guidelines are translated into protocols, which give a contextual description of medical practice provided in a given region. Possibly, protocols developed by regional multidisciplinary maternity care networks (MCNs) contribute to the regional variation in IOL.

### Aims and objectives

The aim of this study was to assess the variation between regional protocols and national guidelines regarding recommendations on IOL and the extent to which this contributes to practice variation.

### Method

We performed a systematic document analysis using the Ready materials, Extract data, Analyze, Distil (READ) approach. National guidelines (n = 4) and regional protocols (n = 18) from six MCNs on topics linked to IOL were assessed between October 2021 and April 2022. An analytical framework was used to extract data for the comparison of regional protocols.

### Results

Some MCNs followed all the recommendations of national guidelines in their regional protocols, others developed their own recommendations, and for some this varied per topic. When developing their own recommendations, MCNs with a high percentage of IOL added additional risk factors and stricter cut-off values. In contrast, MCNs with a low percentage of IOL added more care options for continuing midwife-led care. No clear relationship was observed between the Appraisal of Guidelines for Research & Evaluation (AGREE) scores

**Data Availability Statement:** National guidelines referenced in this study are publicly available at https://www.nvog.nl/ and https://richtlijnendatabase.nl/. The regional protocols for

six Maternity Care Networks can be accessed via the public websites of the respective Maternity Care Networks. Two Maternity Care Networks have chosen not to make their regional protocols publicly accessible. Access to these protocols requires permission from the boards of the respective Maternity Care Networks. Data requests for Maternity Care Network 1 and 2, and general data access requests can be arranged through the institutional point of contact, Dr. M. Hendrix (marijke.hendrix@zuyd.nl). The assessments of national guidelines using the AGREE II tool, along with anonymized frameworks for all regional protocols, will be made available upon reasonable request.

**Funding:** This study was supported by a grant from the Netherlands Organisation for Health Research and Development (www.ZonMw.nl) grant no. 543003312 for the VALID-study paid to the Research Centre for Midwifery Science and a personal PhD Scholarship awarded to DCZ granted by the Royal Dutch Organization of Midwives (KNOV; www.knov.nl). The funders had no role in study design, data collection and analysis, decision to publish, or preparation of the manuscript.

**Competing interests:** The authors have declared that no competing interests exist.

of the national guidelines and the extent to which regional protocols complied with the recommendations.

## Conclusion

The translation of national guidelines to regional protocols seemed arbitrary and not very systematic. To reduce unwarranted practice variation in the use of IOL, guidance is needed to better align regional protocols with national guidelines, while including appropriate contextual factors and allowing women's preferences. Additionally, healthcare providers should be trained in practicing evidence-based medicine instead of using evidence.

## Introduction

Practice variation in health care is gaining attention as a topic in research. Variation in medical practice has been described since the 1930s and variations are seen in diagnoses, contact frequencies, referral rates to more specialized care, and the number of interventions [1]. Variation in itself is not remarkable, because medical conditions and patients' preferences vary. If variation cannot be explained by medical conditions or patient preferences, and there are compelling evidence-based recommendations, practice variation is unwarranted [2, 3]. Unwarranted practice variation is potentially harmful because it can lead to underuse or overuse of interventions, unequal access to good quality care, and higher healthcare costs [2, 4].

To understand the causes of medical practice variation, a sociological model of practice variation has been developed [1, 5]. This model distinguishes three levels 1) micro-level: mechanisms that influence the interaction and decision-making process between the healthcare provider and the patient, such as the provider's attitude and self-efficacy and the patient's attitude and preferences, 2) meso-level: mechanisms that influence practices and organizations, such as regional protocols and regional culture, and 3) macro-level: mechanisms at the (inter) national levels, such as national guidelines and health care systems. Limiting practice variation was one of the reasons why health professionals began to develop clinical practice guidelines. The first guidelines included recommendations based on expert consensus [6, 7]. Later, evidence-based medicine (EBM) was introduced as a counter-movement to authority-based medicine [8]. In EBM, a clinical decision for each individual patient is made by integrating clinical expertise with the best available evidence and the patient preferences [8]. The AGREE Collaboration recognized the importance of high-quality evidence-based guidelines internationally and developed the AGREE instrument to assess the quality of clinical practice guidelines [9]. A systematic literature search is the basis of a guideline, this combined with clinical expertise, patient preferences, and consideration of cost-effectiveness, results in recommendations for optimal care for patients and care providers [10, 11].

Until recently, guidelines for maternity care in the Netherlands were mainly developed monodisciplinary by the professional organizations of obstetricians, midwives, and pediatricians separately [12]. More and more, these guidelines are being developed on a multidisciplinary basis using standardized procedures described by the Dutch Federation of medical specialists [12]. Multidisciplinary guideline development is important because several disciplines are involved in maternity care and often provide care for the same pregnant woman (Box 1). Regionally, primary care midwives, obstetricians, and other disciplines such as pediatricians and maternity care assistants collaborate in maternity care networks (MCNs) [13, 14]. Collaboration in MCNs has intensified over the last decade and has stimulated the

## Box 1. Maternity care in the Netherlands.

In the Netherlands, primary care midwives provide care to women with a low-risk pregnancy as independent healthcare professionals and are able to make autonomous decisions with the woman about childbirth interventions or referral to obstetrician-led care [37]. Indications for a referral from midwife-led to obstetrician-led care are described in the obstetric indication list of 2003 and in national multidisciplinary guidelines [38]. In obstetrician-led care, hospital-based midwives and obstetricians provide care after referral and can provide childbirth interventions such as augmentation of labor, analgesia, and instrumental birth [37].

Primary care midwives, obstetricians, and other disciplines such as pediatricians and maternity care assistants collaborate regionally in maternity care networks (MCNs) [13, 14]. An MCN is usually situated around one hospital and the midwifery practices in the same region. The number of professionals involved varies from about 30 to 120, depending on the number of births and the level of urbanization in the region. Professionals in an MCN are collectively responsible for the quality of maternity care in that region and are expected to continually evaluate perinatal outcomes and women's experiences in order to improve the quality and efficiency of their care [14].

development of regional protocols within MCNs [14]. In general, protocols are more context-specific than guidelines and describe the 'who', 'what', 'when', and 'how' of medical practice provided in a given region.

Previous research on regional protocols for perinatal care in Dutch hospitals showed a lack of standardization [15–17]. Not all hospitals had protocols on similar topics, and there was a wide variation in developmental methodology and content. It seemed unclear to health professionals what the purpose and content of a protocol should be and how to formulate recommendations that reflect the quality of the supporting evidence [15, 17]. There is also no clarity on how regional protocols relate to national guidelines. Development of clinical practice guidelines was started to reduce practice variation and methodological clarity exists on how to develop them in accordance with the AGREE recommendations, however, no guidance exists on the development procedure or content of regional protocols. The lack of guidance makes it possible that regional protocols are a potential factor for practice variation.

In this study, we focused on guidelines and protocols with recommendations on induction of labor (IOL). IOL is an intervention in perinatal care that is used for various indications [18]. Despite national guidelines recommending situations in which IOL is appropriate care, there are large regional differences in the use of this intervention in the Netherlands, with percentages ranging from 14.3% to 41.1% [19]. Because IOL is a major intervention during pregnancy with the potential for harm, it should only be performed on medical indication [20]. In situations complicated by pre-eclampsia of diabetes mellitus, the benefits of IOL for mother and child outweigh the harms. However, IOL is also associated with less favourable outcomes such as the risk of uterine hyperstimulation and rupture, fetal distress, and more unplanned caesarean sections [20]. This study focused on variation in regional protocols as one of the factors that may contribute to practice variation at the meso-level of the sociological model of practice variation. The purpose of this study was to analyze variation between regional protocols, and variation between regional protocols and national guidelines regarding recommendations for

IOL. Additionally, we explored the extent to which national guidelines were used in regional protocols and whether this was related to the quality of the national guidelines.

## Methods

### 2.1 Design

We conducted a critical document analysis to gain insight into and understanding of the regional protocols and national guidelines. We used the READ approach as a systematic qualitative description approach [21]. The READ (Ready materials, Extract data, Analyze data, Distil) approach provides a step-by-step guide to document analysis in health policy research, extracting insight from documents, while ensuring rigor in the analysis. The READ approach can be adapted to different purposes and types of research and is useful for understanding policy content at regional, national, or global level.

### 2.2 Setting

This study was part of the larger VALID study, which describes practice variation in IOL between MCNs in the Netherlands and explores the different mechanisms that influence the decision-making process about IOL. The aim was to select a total of six MCNs for the VALID study, three MCNs with a high percentage of IOL and three MCNs with a low percentage of IOL. In the Netherlands, the Perined database includes data from medical records of almost all births [19]. For the VALID-study, the records with a relatively low risk for severe pregnancy complications in the years 2016–2018 were selected. IOL rates in these groups were calculated per MCN with case-mix correction for available socio-demographic factors. The six MCNs with the highest percentage of IOL and the six VSVs with the lowest percentage of IOL were approached for participation. In both groups, at least three MCNs were willing to participate, with the final selection taking into account geographical distribution. Both groups also included an MCN situated around an academic hospital.

### 2.3 National guidelines

**2.3.1 Search and selection.** In October 2021, we searched for relevant national guidelines in the Dutch guideline database and on the websites of the Dutch associations of obstetricians and midwives. Relevant guidelines were all national guidelines on maternity care related issues that described specific recommendations for IOL in common situations. From this selection, the following four topics were identified: 1) the management of shoulder dystocia (including recommendations for suspected macrosomia or large for gestational age infants), 2) reduced fetal movement, 3) elective induction of labor, and 4) late term pregnancy (≥41 weeks) (Table 1).

Table 1. Subject, authorization parties, and publication date of the included national guidelines.

| Subject | Authorization parties | Publication date |
|---|---|---|
| Shoulder dystocia | Dutch association of obstetricians | 17-09-2008 |
| Reduced fetal movements | Dutch association of obstetricians<br>Dutch association of midwives | 12–2013 |
| Elective induction of labour | Dutch association of obstetricians | 15-4-2020 |
| Late term pregnancy<br>(≥41 weeks pregnancy) | Dutch association of obstetricians<br>Dutch association of midwives<br>Dutch association of paediatricians<br>Client organizations | 15-2-2021 |

**2.3.2 Quality assessment.** The quality of the national guidelines was assessed using the AGREE II instrument [22]. This instrument consists of two overall items and 23 items divided into six domains (Box 2), which assess the quality and development process of the guidelines. Each item, except the two overall items, is scored on a 7-point scale (1—strongly disagree to 7—strongly agree). A sum score and percentage is calculated for each domain. One of the overall items asks whether the guideline can be used in practice (yes—no) and the other item asks for a numerical quality score. Each guideline was scored independently by three of the authors, individual scores were compared and consensus was reached after discussion. In addition, we extracted the recommendations for IOL from each of the four guidelines.

Box 2. Six domains of AGREE II.

Domain 1: Scope and purpose

Domain 2: Stakeholder involvement

Domain 3: Rigor of development

Domain 4: Clarity of presentation

Domain 5: Applicability

Domain 6: Editorial independence

## 2.4 Regional protocols

**2.4.1 Search and selection.** Regional protocols from the six participating MCNs were collected for in-depth content analysis and assessment. They were collected between 1 October 2021 and 15 April 2022 from the MCN websites and by contacting the MCNs. Regional protocols were eligible if they described recommended care related to IOL for the topics shoulder dystocia, large-for-gestational-age or macrosomia, reduced fetal movements, elective induction of labor, or late term pregnancy (≥41 weeks).

**2.4.2 Analytical framework.** Because the AGREE II instrument was developed specifically for guidelines, it contains elements that are applicable to national guidelines. Regional protocols have a different structure and include different elements, which makes the AGREE II instrument not suitable for analyzing regional protocols. An analytical framework (S1 Appendix. Analytical framework for analyzing regional protocols.) was developed for this study, based on the domains of the AGREE II instrument [22] and complemented with items for critical document analysis [21]. A first version was pilot tested on the regional protocols for late term pregnancy and adjusted for data extraction to meet the purpose of this study. The analytical framework consisted of questions about the development procedure such as scope and target population. Other questions focused on the content of the regional protocols, with questions about relevant recommendations, supporting evidence for the recommendations, clarity of presentation, applicability, and general impression. The first author answered the questions of the analytical framework for each protocol based on the information given in the

protocol. Subsequently, the second author monitored the answers given and these were discussed together for the final assessment.

## 2.5 Analysis

The collected data resulted in an extensive dataset, which we analyzed by systematically answering the questions of the analytical framework for each MCN and reasoning what effect the described care might have on IOL rates. The different MCNs were then compared to see if the outcomes differed and if there was a relationship between the outcomes of the analysis and whether the MCN had a high or low IOL rate. In accordance with the READ method, we also took a holistic view of all documents to see what variation there was within and between the documents [21]. The scores of the AGREE II instrument were compared with the recommendations of the regional protocols to see if there was a relationship between the score and the extent to which national guidelines were used in the regional protocols. To increase the validity of our findings, we discussed them in detail with the research team and checked them in the original data before drawing conclusions.

## 2.6 Ethics

According to the 'Act governing research involving human subjects' in The Netherlands (WMO), formal ethical approval by a research ethics committee is only required for medical research where participants are subject to interventions or procedures, or are required to follow specific, research related rules of behavior [23]. Because this research is a document analysis, none of these apply. As the focus of the study was on regional protocols and national guidelines, there was no need for anonymization of documents and no written consent was required.

## Results

### 3.1 Quality assessment national guidelines

In the Netherlands, a standardized method for developing multidisciplinary national guidelines was introduced in 2012. Emphasis was put on an extensive literature review and reporting all individuals and stakeholders involved in the development procedure [12]. The AGREE II domain scores of the analyzed guidelines reflected this process of standardization: the newest guideline (late term pregnancy; 2021) showed the highest scores and the eldest guideline (shoulder dystocia; 2008) had the lowest scores (Table 2). The second newest guideline 'Elective induction of labor' (2020) had a remarkably low score on '*Stakeholder involvement*' (domain 2). The development process of this guideline was monodisciplinary, three other disciplines were only involved in the external review and authorization of the final version.

The overall scores showed the highest scores for the guidelines on late term pregnancy (5.33) and reduced fetal movements (4.57). All reviewing authors recognized both guidelines as appropriate for use in clinical practice, according to the question in the AGREE II instrument. In contrast, the guidelines on elective induction of labor and shoulder dystocia had low overall scores (resp. 3.67 and 2.33) and were not recognized by the reviewing authors as appropriate for use in clinical practice.

### 3.2 General impression regional protocols

The purpose of all regional protocols was to describe maternal and perinatal care in the MCN, including primary midwife-led care and secondary obstetrician-led care. Not every MCN had a regional protocol for all four selected topics. None of the MCNs had a protocol for elective

**Table 2. Quality assessment of four national guidelines based on the AGREE II instrument.**

| Guideline (year of publication) | D1 | D2 | D3 | D4 | D5 | D6 | Quality score (0–7) | Use |
|---|---|---|---|---|---|---|---|---|
| Shoulder dystocia (2008) | 7.4% | 11.1% | 23.6% | 57.4% | 5.6% | 0.0% | 2.33 | No |
| Reduced fetal movements (2013) | 96.3% | 64.8% | 41.0% | 81.5% | 30.6% | 0.0% | 4.57 | Yes |
| Elective induction of labor (2020) | 68.5% | 25.9% | 66.0% | 85.2% | 33.3% | 44.4% | 3.67 | No |
| Late term pregnancy (≥41 weeks pregnancy) (2021) | 61.1% | 66.7% | 80.6% | 94.4% | 33.3% | 100% | 5.33 | Yes |

Based on domain scores:

D1 = Domain 1: Scope and purpose

D2 = Domain 2: Stakeholder involvement

D3 = Domain 3: Rigor of development

D4 = Domain 4: Clarity of presentation

D5 = Domain 5: Applicability

D6 = Domain 6: Editorial independence

Overall items:

Quality score = numerical score of the guideline rated by the reviewing authors

Use = verdict of the reviewing authors if they recognize the guideline as appropriate for use in clinical practice

induction of labor, and for the topics late term pregnancy and macrosomia two or three protocols were identified in one MCN. In total, 18 regional protocols were identified in the six participating MCNs describing the recommended care for (prevention of) shoulder dystocia, large-for-gestational-age or macrosomia (n = 6), reduced fetal movements (n = 5), and late term pregnancy (≥41 weeks) (n = 7) (Table 3).

A large variation was seen in document types. Documents varied from short and staccato protocols of one page describing what to do, to extensive protocols of ten pages describing not only recommended care, but also providing background, flowcharts, and recommendations for counseling. The documents were named differently, such as protocol, guideline, or care pathway. Three of the six MCNs consequently described authors, version number, and revision date in the document, while the other three MCNs did not.

Some regional protocols described the full range of perinatal care in the specific situation, including a description of what should be done in the event of a referral from midwife-led care to obstetrician-led care for all professionals involved. Others focused separately on recommended midwife-led primary care or obstetrician-led secondary care.

### 3.3 Development procedure regional protocols

Five of the six MCNs described a comparable development procedure. Their regional protocols were developed by a multidisciplinary panel of healthcare providers. Subsequently, a draft version was presented for feedback to their colleagues. In four out of five MCNs, all colleagues

**Table 3. Amount of regional protocols per subject (n = 18).**

| Subject | MCN 1 | MCN 2 | MCN 3 | MCN 4 | MCN 5 | MCN 6 |
|---|---|---|---|---|---|---|
| Shoulder dystocia | 1 | 1 | 0 | 1 | 0 | 3 |
| Reduced fetal movements | 0 | 1 | 1 | 1 | 1 | 1 |
| Elective induction of labor | 0 | 0 | 0 | 0 | 0 | 0 |
| Late term pregnancy (≥41weeks) | 1 | 2 | 1 | 2 | 0 | 1 |

were consulted to approve the final protocol. In one other MCN approval was done by a specially mandated committee. The sixth MCN described a different procedure. A mandated multidisciplinary workgroup formulated specific statements on care based on a review of national and international guidelines and other relevant literature. The other care professionals in this MCN were subsequently asked to react to these statements, and based on these reactions the final recommendations were formulated by the working group and published. None of the MCNs described the participation of women in the development procedure of a regional protocol.

## 3.4 Recommendations in regional protocols

We observed variations between MCNs in the use of national guidelines in their protocols. However, we did not observe a clear relationship between the extent to which national guidelines were used in regional protocols and the overall AGREE II score of the national guidelines. Two MCNs used the national guidelines as the main source for their protocols and included recommendations from the national guideline in their regional protocols (Table 4). The protocols of these two MCNs followed the national guidelines quite precisely. One of these MCNs belonged to the high IOL group and the other to the low IOL group. Two other MCNs, also one high and one low IOL MCN, developed regional protocols and formulated recommendations based on self-collected evidence, such as data from the Dutch national perinatal register, individual studies, documents from the Dutch Association of Midwives, and international and national guidelines. The last two MCNs used both these strategies for developing protocols, depending on the topic.

Self-developed recommendations varied in several ways. Based on the variation in these recommendations, a possible relationship between the regional protocols and a high or low percentage of IOL appeared. Firstly, protocols from MCNs with a high percentage of IOLs described additional risk factors compared to national guidelines, often expanding the group eligible for IOLs. Factors such as advanced maternal age, smoking, or maternal body mass index $>40$ kg/m$^2$ were described as indicators to induce labor at 41 weeks gestation instead of considering expectant management. Secondly, cut-off values or definitions were defined differently in MCNs compared to national guidelines. For example, some protocols of MCNs with a high percentage of IOL recommended IOL in cases of suspected fetal macrosomia, based on a specific and rather strict cut-off value at the fetal growth scan (f.e. the 75$^{th}$

**Table 4. Are the recommendations in the existing regional protocols in line with the national guideline?.**

| Subject* | MCN 1–high % IOL | MCN 2–high % IOL | MCN 3–high % IOL | MCN 4–low % IOL | MCN 5–low % IOL | MCN 6–low % IOL |
|---|---|---|---|---|---|---|
| **Shoulder dystocia** *AGREE overall score: 2.33* | No | Yes | Yes[#] | No | Yes[#] | No |
| **Reduced fetal movements** *AGREE overall score: 4.57* | No protocol available | Yes | No | No | Yes | No |
| **Late term pregnancy** *AGREE overall score: 5.33* | No | Yes—recommendations in line with the old guideline 2007 | No | Yes | Yes[#] –recommendations in line with the old guideline 2007 | No |

* no regional protocols present on the subject 'Elective induction of labor

[#] For this MCN no regional protocol existed, they referenced to the national guideline

percentile). Other MCNs with a low percentage of IOL stated no specific cut-off value at the growth scan for the management of suspected fetal macrosomia and indicated continuing midwife-led care until referral was considered necessary by the primary care midwife. Thirdly, additional care options were described compared to care described in the national guidelines. Examples in MCNs with a low percentage of IOLs were the option of artificial rupture of membranes in midwife-led care to induce labor and the option of antenatal cardiotocography for fetal assessment in midwife-led care in cases of reduced fetal movements instead of cardiotocography in obstetrician-led care. These options create opportunities for more continuity of care and potentially fewer referrals for IOL in obstetrician-led care. Extra diagnostic tests such as ultrasound or oral glucose testing were described as additional options in MCNs with a higher percentage of IOL, potentially leading to more situations in which IOL is recommended. Self-developed recommendations had mostly no references to scientific literature and were not explained with considerations of care providers.

### 3.5 Other observations

Regional protocols varied in the way they described different care options for women and the involvement of women in the final decision about care options, including IOL. Some regional protocols did not mention women in their protocols, and other regional protocols, all from MCNs with low percentages of IOL, described women as the final decision maker. These protocols explicitly described that woman's preferences should be explored and that the woman should be allowed to decide on treatment.

The provision of information was most frequently described in protocols on reduced fetal movements. Protocols from MCNs with high and low percentages of IOL were equally likely to mention 'personalized care' and 'treatment based on counseling', without further specification of the content of this care. Recommendations for counseling were regularly described in protocols for late term pregnancy and large-for-gestational age. However, the protocols were limited in their specifications.

We observed differences in the writing style used in the MCNs. In one MCN, a prescriptive writing style was observed, including strict cut-off values for interventions and prescriptive instructions on what to do if a woman requested care different from the regional protocol.

## Discussion

In this critical document analysis, we found a large variation between regional protocols, which suggests that regional protocols may contribute to the current practice variation in IOL in the Netherlands. Some MCNs followed the recommendations of the national guidelines for all topics, other MCNs developed their own recommendations, and some MCNs used both these strategies depending on the topic. When developing their own recommendations, MCNs added additional risk factors, care options, and specific cut-off values. It appears that MCNs with a low percentage of IOL were more likely to describe additional options where women could stay in midwife-led primary care and to describe the involvement of women in decision-making. MCNs with a high percentage of IOL seemed to describe more indications for interventions such as ultrasound or fetal monitoring, and more indications or stricter cut-off values for IOL compared with MCNS with a low percentage of IOL. Causality could not be proven on the basis of this study, but these results suggest that regional protocols from regions with a high percentage of IOL indicate more situations in which IOL is recommended. Both groups were equally likely to mention 'personalized care' and 'treatment based on counseling'. No clear relationship was observed between the AGREE scores of the national guidelines and the extent to which the regional protocols complied with national recommendations.

## National guidelines and regional protocols

Several sources of unwarranted practice variation have been described in the literature. One of the most important sources of unwarranted variation appears to be misinterpretation or misapplication of relevant clinical evidence [24]. One strategy to minimize this type of unwarranted variation is the development of national guidelines [12, 25]. National guideline development provides an opportunity to ensure that sufficient knowledge and resources are available. Guideline development is a specialized task that requires experts who have the time to conduct a systematic analysis of the literature and formulate evidence-based recommendations [26]. According to our study, national guidelines do not always seem to reflect current professional practice. Several national guidelines were out of date and do not support healthcare providers with the latest evidence and current issues. Regional protocols do not appear to adequately compensate for these shortcomings. Therefore, national associations need to ensure that guidelines are regularly reviewed to determine whether they are still up-to-date or need to be revised [12]. In this way, guidelines can truly be the evidence base for clinical decision-making.

Our analysis confirms previous research that there is great variation between regional protocols [15, 17]. We found that evidence from national guidelines is not always incorporated into regional protocols, and some MCNs redo the guideline development process to develop a regional protocol. Considering this, there is a need for clarity in Dutch maternity care about the relationship between national guidelines and regional protocols, and about the content of regional protocols. Ultimately, a national guideline is the leading document that provides the systematic literature analysis and evidence-based recommendations on when certain care is needed. Guidelines should also describe where the evidence is less strong and where there is room for women's preferences. As a result, guidelines give direction to regional protocols, leaving only context-specific factors such as 'who' and 'how' to be specified [26]. This can avoid random use of evidence in regional protocols and allows for women's preferences, potentially reducing regional practice variation between MCNs. If recommendations in regional protocols do differ from those in the national guideline or, for example, if different cut-off values are used, this should be justified in the protocol.

## The risk of over-standardization

Regional protocols provide an opportunity for a concrete description of medical practice provided in a particular region. This standardization in protocols can help to reduce healthcare provider subjectivity, bias, and uncertainty, thereby reducing unwarranted practice variation [26, 27]. However, while protocols should facilitate evidence-based practice, they should not promote undesirable standardization of care. Evidence-based practice combines knowledge of the patient's clinical condition, with the scientific literature, and the patient's individual preferences. The risk of undesirable standardization is that these factors are not sufficiently taken into account [27]. National guidelines can provide the scientific literature for regional protocols, and professional expertise is important in addition to knowledge of the client's clinical condition and previous practices. Healthcare providers need to be aware of their directing role in medical practice performed. It appears healthcare providers also need support in interpreting evidence and in shared decision-making [28, 29]. For healthcare providers, guidelines and guideline developers represent a new authority that makes evidence available for use and makes recommendations that must be followed. As a result, healthcare providers become 'evidence users', rather than reviewing and applying the available evidence themselves and developing a (self)critical attitude [30].

Too much standardization in regional protocols through additional risk factors and strict cut-off values may result in different management of certain obstetric problems in neighboring MCNs with similar populations. This can also be a reason for practice variation between different MCNs. Research has shown that the clinical decision-making of healthcare providers is influenced by several factors, such as geographical and organizational factors [1, 31]. Over-standardization of protocols can make it difficult to align protocols across regions. It can also lead to confusion among healthcare professionals providing care in different regions, possibly leading to more unwarranted practice variation.

## Patient preferences

One factor in good evidence-based practice is whether patient's individual preferences are taken into account [12, 26]. Shared decision-making is fundamental to maternity care and is a collaborative process between the healthcare professional and the patient to make healthcare decisions using respectful communication [26, 32]. Our analysis showed that the role of women's choice in clinical decision-making was not always described in regional protocols. Counseling seemed to be described in cases where there is a lack of good evidence, or where the available evidence gives room for equivalent care options. Regions with a lower percentage of IOL appeared to be more likely to describe women having a choice or making the final decision. This may indicate that these regions are more attentive towards involving women in the decision-making process. Previous research has shown that most women receiving maternity care prefer physiological labor and birth [33]. Increasing women's involvement in clinical decision-making could therefore lead to less medicalization. Following patients' preferences is a source of justified variation and appears to reduce practice variation between hospitals and simultaneously increase variation within hospitals [34]. Women's preferences should be described as a factor to be considered in clinical decision-making as a standard in regional protocols. Attention should also be given to evaluating how these preferences are managed in practice and whether healthcare professionals have sufficient skills to support women's active participation and involvement in maternity care.

## Strengths and limitations

This study is an exploration of regional protocols based on four topics for IOL in six MCNs. Although this is a relatively small sample, we saw a large variation in regional protocols. It is likely that protocols on other topics show similar variation. At the time of the analysis, some national guidelines had recently been published or were under revision. It is possible that these national guidelines have been partially implemented in the regional protocols, but this could not be traced due to the limited use of references to the evidence base in the regional protocols.

As we did not observe the actual implementation of the regional protocols in practice, we cannot conclude to what extent clinical decision-making is in line with the regional protocols. However, the differences in recommendations in the regional protocols and the apparent relationship with high and low percentage of IOL in MCNs makes it plausible that the regional protocols are a contributing factor to practice variation.

Several techniques have been used to enhance qualitative rigor, such as the use of a systematic method, peer debriefing, and the use of multiple researchers in the study (investor triangulation). Data triangulation was achieved by searching for documents on websites, online databases, and requests to the MCN [35, 36].

## Conclusion

Overall, this study shows that there is variation in regional protocols, which can lead to unwanted standardization and unwarranted practice variation. There is a need for guidance on the development and content of regional protocols. In order to better align regional protocols with national guidelines, while adequately describing contextual factors, and making room for women's preferences and professional expertise. Healthcare providers should be trained in the use of evidence and in shared decision-making to become evidence-based practitioners. At the national level, there is a need to work on up-to-date guidelines and to communicate the relationship between national guidelines and regional protocols. Furthermore, research is needed into other mechanisms that contribute to practice variation.

## Supporting information

**S1 Appendix. Analytical framework.**
(DOCX)

## Author Contributions

**Conceptualization:** Dirkje C. Zondag, Pien M. Offerhaus, Judit K. J. Keulen, Tamar M. van Haaren–ten Haken, Marianne J. Nieuwenhuijze.

**Data curation:** Dirkje C. Zondag, Pien M. Offerhaus, Judit K. J. Keulen.

**Formal analysis:** Dirkje C. Zondag, Pien M. Offerhaus, Judit K. J. Keulen.

**Funding acquisition:** Dirkje C. Zondag, Pien M. Offerhaus, Tamar M. van Haaren–ten Haken, Marianne J. Nieuwenhuijze.

**Investigation:** Dirkje C. Zondag.

**Methodology:** Dirkje C. Zondag, Pien M. Offerhaus, Judit K. J. Keulen.

**Project administration:** Dirkje C. Zondag.

**Resources:** Dirkje C. Zondag.

**Supervision:** Marianne J. Nieuwenhuijze.

**Validation:** Dirkje C. Zondag, Pien M. Offerhaus, Judit K. J. Keulen, Tamar M. van Haaren–ten Haken, Marianne J. Nieuwenhuijze.

**Visualization:** Dirkje C. Zondag.

**Writing – original draft:** Dirkje C. Zondag.

**Writing – review & editing:** Dirkje C. Zondag, Pien M. Offerhaus, Judit K. J. Keulen, Tamar M. van Haaren–ten Haken, Marianne J. Nieuwenhuijze.

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
