## [Decision Letter · Decision Letter 0]

2 Jan 2024

PONE-D-23-36626Practice variation in induction of labor: a critical document analysis on the contribution of regional protocolsPLOS ONE

Dear Dr. Zondag,

Thank you for submitting your manuscript to PLOS ONE. After careful consideration, we feel that it has merit but does not fully meet PLOS ONE’s publication criteria as it currently stands. Therefore, we invite you to submit a revised version of the manuscript that addresses the points raised during the review process.

We look forward to receiving your revised manuscript.

Kind regards,

Mubarick Nungbaso Asumah, MPhil, Bsc

Academic Editor

PLOS ONE

Journal Requirements:

Reviewers' comments:

Reviewer's Responses to Questions

**Comments to the Author**

1. Is the manuscript technically sound, and do the data support the conclusions?

Reviewer #1: Yes

Reviewer #2: Yes

2. Has the statistical analysis been performed appropriately and rigorously? 

Reviewer #1: N/A

Reviewer #2: Yes

3. Have the authors made all data underlying the findings in their manuscript fully available?

Reviewer #1: No

Reviewer #2: No

4. Is the manuscript presented in an intelligible fashion and written in standard English?

Reviewer #1: Yes

Reviewer #2: Yes

5. Review Comments to the Author

Reviewer #1: Thank you for the opportunity to review this manuscript. I found it to be technically sound and clearly written.

A couple of issues I think the authors might want to consider: include more examples of how regional protocols differ from national guidelines; spell out to the reader what the consequences are of unwarranted practice variation (e.g. how might pregnant people and their babies be more or less at risk due to practice variation).

The results section describes two of the national guidelines as being judged not appropriate for use in clinical practice; are these being revised and updated? While I recognise that this issue is beyond the scope of the manuscript, it might be worth mentioning in the discussion section something about these two guidelines. Do the regional protocols compensate in any way for the shortcomings of these guidelines, i.e. are the regional protocols more robust or better quality?

With regard to the abstract, I would like to see the READ and AGREE acronyms explained and some clarification of the final sentence in the conclusion of the abstract would be beneficial.

Regarding language, in some parts of the manuscript it seems that pregnant people are referred to as patients; I think it is better to avoid this language and use terms such as pregnant people/individuals instead.

I answered 'no' to the question 'Have the authors made all data underlying the findings in their manuscript fully available?' because I was expecting to see at least an example of how the questions in the supplementary table were answered. It is also not clear to me how the percentages were arrived at in Table 2 'Quality assessment of four national guidelines based on the AGREE II instrument'.

Reviewer #2: This important piece of research highlights the extent to which national guidelines and local protocols/guidelines vary. This is an international as well as national issue.

The aim of the study was to assess the variation between regional guidelines/protocols when compared to national guidelines for specific obstetric conditions (shoulder dystocia, reduced fetal movements, late term pregnancy >/= 41 weeks’ gestation) for which induction of labour is a potential outcome and the extent to which this contributes to practice variation. The study is part of the VALID (VAriation in Labour InDuction) study, which describes practice variation in induction of labour between 77 maternity care networks in the Netherlands.

The researchers selected 6 maternity care networks (MCNs) for the current study, 3 of which had a high percentage of induction of labour (IOL) and 3 of which had a low percentage IOL. The researchers have not elucidated how these 6 MCNs were chosen, was it random selection in each group? the 3 with the highest percentage of IOL in each group? the 3 with the lowest percentage of IOL in each group?

The researchers performed the READ approach (ready materials, extract data, analyse data, distil findings) for document analysis.

The quality of the national guidelines was assessed using the AGREE (Appraisal of Guidelines for Research and Evaluation) II instrument. The researchers developed an analytical framework based on the domains of the AGREE II instrument to appraise the regional protocols/guidelines.

Were the MCNs with the highest percentage of IOL located in areas that predominantly cared for women with high risk pregnancies and those with the lowest percentage of IOL in areas that predominantly cared for women with low risk pregnancies?

Ethics statement Lines 133-139 would make more sense at the end of the methods 2.6 Ethics

Line 144 for changed to from the VALID study

Move Table 1 to after the heading for Table 1

Late term or late-term should be standardised throughout the text/tables

All instances of 41 weeks should be standardised to >/= 41 weeks

Reference is for AGREE II instrument but AGREE is used throughout text

Line 163 change box 1 to box 2

Line 302 change to Firstly

Line 306 change to Secondly

6. PLOS authors have the option to publish the peer review history of their article (what does this mean?). If published, this will include your full peer review and any attached files.

Reviewer #1: **Yes: **Fiona Stewart

Reviewer #2: No

---

## [Author Response · Author response to Decision Letter 0]

16 Feb 2024

Dear M. Nungbaso Asumah, 

Thank you for the feedback and guidance on how to meet the journal’s requirements. We have updated the manuscript in order to meet the PLOS ONE’s style requirements. The data sharing plan has been discussed within the research team. It was agreed that the data can be made available after the manuscript is accepted for publication. The assessments of the national guidelines using the AGREE II tool and the completed frameworks for all regional protocols were anonymised and can be made available on request. In addition, we have included a caption for the supporting information file at the end of the manuscript and updated in-text references. We have added one reference to the manuscript, and changed the references and reference list accordingly. We have not retracted any references. 

We have responded to the feedback of both reviewers in the text below. The line numbers refer to the lines in the manuscript without track and changes. Thank you for considering our work for publication in PLOS ONE. 

Review 1: 

Thank you for the opportunity to review this manuscript. I found it to be technically sound and clearly written.

Many thanks for your positive feedback. We have read your suggestions with great interest, and respond to them below.

A couple of issues I think the authors might want to consider: include more examples of how regional protocols differ from national guidelines; spell out to the reader what the consequences are of unwarranted practice variation (e.g. how might pregnant people and their babies be more or less at risk due to practice variation).

Thank you for asking more information. Regarding the first comment to include more examples of how regional protocols differ from national guidelines, we have changed the text from line 305 onwards. There are several examples in this passage, but we clarified that the examples given refer to differences between regional protocols compared with national guidelines: 

Lines 307-309: ‘Firstly, protocols from MCNs with a high percentage of IOLs described additional risk factors compared to national guidelines, often expanding the group eligible for IOLs. 

Lines 311-319: Secondly, cut-off values or definitions were defined differently in MCNs compared to national guidelines. For example, some protocols of MCNs with a high percentage of IOL recommended IOL in cases of suspected fetal macrosomia, based on a specific and rather strict cut-off value at the fetal growth scan (f.e. 75th percentile). Other MCNs with a low percentage of IOL stated no specific cut-off value at the growth scan for the management of suspected fetal macrosomia and indicated continuing midwife-led care until referral was considered necessary by the primary care midwife. Thirdly, additional care options were described compared to care described in the national guidelines.’ 

In the introduction (lines 105-107), we have added the following information about consequences of unwarranted practice variation in maternity care: ‘Because IOL is a major intervention during pregnancy with the potential for harm, it should only be performed on medical indication [20]. In situations complicated by pre-eclampsia of diabetes mellitus, the benefits of IOL for mother and child outweigh the harms. However, IOL is also associated with less favourable outcomes such as the risk of uterine hyperstimulation and rupture, fetal distress, and more unplanned caesarean sections [20].’

Reference [20]: World Health Organization. WHO recommendations for induction of labour. Geneva: World Health Organization; 2011

The results section describes two of the national guidelines as being judged not appropriate for use in clinical practice; are these being revised and updated? While I recognise that this issue is beyond the scope of the manuscript, it might be worth mentioning in the discussion section something about these two guidelines. Do the regional protocols compensate in any way for the shortcomings of these guidelines, i.e. are the regional protocols more robust or better quality?

Thank you for raising this issue. Of the two national guidelines that are not considered suitable for use in clinical practice, the guideline on Shoulderdystocia is currently being revised. The regional protocols on Shoulderdystocia and large-for-gestational age mostly described concrete recommendations for practice. However, the regional protocols did not include a systematic review of the literature. As a result, the quality of the evidence on which the recommendations are based remains unclear.

The guideline on Elective induction of labour was published in 2020 and not yet subject to revision. Based on the document analysis, there does not seem to be a need for regionally alignment on this topic as there are no protocols for it. 

We added the following in the discussion (lines 377-380): ‘Regional protocols do not appear to adequately compensate for these shortcomings. Therefore, national associations need to ensure that guidelines are regularly reviewed to determine whether they are still up-to-date or need to be revised.’ 

With regard to the abstract, I would like to see the READ and AGREE acronyms explained and some clarification of the final sentence in the conclusion of the abstract would be beneficial.

We have explained the acronyms READ and AGREE in the abstract. However, due to the word count, it was not possible to clarify the final sentence and we had to shorten the rationale. 

Regarding language, in some parts of the manuscript it seems that pregnant people are referred to as patients; I think it is better to avoid this language and use terms such as pregnant people/individuals instead.

We discussed this comment in the research team. We believe that in the parts where the manuscript describes evidence-based practice or practice variation in general ‘patients’ is the right term to use. In the parts where the text refers to pregnant individuals wording like ‘women’s preferences/ involvement’ is used. 

I answered 'no' to the question 'Have the authors made all data underlying the findings in their manuscript fully available?' because I was expecting to see at least an example of how the questions in the supplementary table were answered. It is also not clear to me how the percentages were arrived at in Table 2 'Quality assessment of four national guidelines based on the AGREE II instrument'.

Thank you for this comment. Given the feedback, information appears to be missing in the method section. We changed the text (lines 158-160) to: ‘Each item, except the two overall items, is scored on a 7-point scale (1 - strongly disagree to 7 - strongly agree). A sum score and percentage is calculated for each domain.’ and (lines 187-190) ‘The first author answered the questions of the analytical framework for each protocol based on the information given in the protocol. Subsequently, the second author monitored the answers given and these were discussed together for the final assessment.’ In addition, the anonymized completed frameworks can be requested after publication.

Review 2

This important piece of research highlights the extent to which national guidelines and local protocols/guidelines vary. This is an international as well as national issue.

The aim of the study was to assess the variation between regional guidelines/protocols when compared to national guidelines for specific obstetric conditions (shoulder dystocia, reduced fetal movements, late term pregnancy >/= 41 weeks’ gestation) for which induction of labour is a potential outcome and the extent to which this contributes to practice variation. The study is part of the VALID (VAriation in Labour InDuction) study, which describes practice variation in induction of labour between 77 maternity care networks in the Netherlands.

The researchers selected 6 maternity care networks (MCNs) for the current study, 3 of which had a high percentage of induction of labour (IOL) and 3 of which had a low percentage IOL. The researchers have not elucidated how these 6 MCNs were chosen, was it random selection in each group? the 3 with the highest percentage of IOL in each group? the 3 with the lowest percentage of IOL in each group?

Thank you for raising this point. We expanded the text for a clearer description of the selection process of the MCNs (lines 131-140): 

‘The aim was to select a total of six MCNs for the VALID study, three MCNs with a high percentage of IOL and three MCNs with a low percentage of IOL. In the Netherlands, the Perined database includes data from medical records of almost all births [19]. For the VALID-study, the records with a relatively low risk for severe pregnancy complications in the years 2016-2018 were selected. IOL rates in these groups were calculated per MCN with case-mix correction for available socio-demographic factors. The six MCNs with the highest percentage of IOL and the six VSVs with the lowest percentage of IOL were approached for participation. In both groups, at least three MCNs were willing to participate. For the final selection we took into account geographical distribution. Both groups also included an MCN situated around an academic hospital.’

The researchers performed the READ approach (ready materials, extract data, analyse data, distil findings) for document analysis.

The quality of the national guidelines was assessed using the AGREE (Appraisal of Guidelines for Research and Evaluation) II instrument. The researchers developed an analytical framework based on the domains of the AGREE II instrument to appraise the regional protocols/guidelines.

Were the MCNs with the highest percentage of IOL located in areas that predominantly cared for women with high risk pregnancies and those with the lowest percentage of IOL in areas that predominantly cared for women with low risk pregnancies?

Thank you for this comment. Each MCNs in the Netherlands provides care for high risk pregnancies and low risk pregnancies. The Dutch maternity care system is further explained in Box 1 in the article. Both the low IOL and the high IOL group also included an academic hospital offering care to specific high risk pregnancies, such as premature births <32 weeks and pregnancies complicated with severe fetal defects. We expanded the text to clarify the inclusion of academic hospitals in both groups MCNs (line 140): ‘Both groups also included an MCN situated around an academic hospital.’ 

Ethics statement Lines 133-139 would make more sense at the end of the methods 2.6 Ethics

We have moved the ethics statement to the end of the methods (lines 205-212) section and made a new subheading ‘Ethics’. 

Line 144 for changed to from the VALID study

We have discussed this comment in the research team and decided not to make the suggested change, because six MCNs were selected for the VALID study and these same MCNs participated in this study. 

Move Table 1 to after the heading for Table 1

Thank you for noticing this. We placed the heading before Table 1.

Late term or late-term should be standardised throughout the text/tables

We have changed the wording to 'late term' throughout the text.

All instances of 41 weeks should be standardised to >/= 41 weeks

Thank you for this comment. We have standardised the used term to ‘≥41 weeks’ throughout the whole article. 

Reference is for AGREE II instrument but AGREE is used throughout text

We changed ‘AGREE’ into ‘AGREE II’ in cases where the instrument was intended. 

Line 163 change box 1 to box 2

We changed ‘box 1’ into ‘box 2’.

Line 302 change to Firstly

We changed ‘First’ into ‘Firstly’.

Line 306 change to Secondly

We changed ‘Second’ into ‘Secondly’.

Kind regards, 

Lianne Zondag, RM MSc

---

## [Editor Report · Decision Letter 1]

12 Sep 2024

Practice variation in induction of labor: a critical document analysis on the contribution of regional protocols

PONE-D-23-36626R1

Dear Dr. Zondag,

We’re pleased to inform you that your manuscript has been judged scientifically suitable for publication and will be formally accepted for publication once it meets all outstanding technical requirements.

Kind regards,

Mubarick Nungbaso Asumah, MPhil, Bsc

Academic Editor

PLOS ONE

Additional Editor Comments (optional):

Although we were unable to secure the same reviewers to assess the authors' responses, I have personally reviewed the responses in detail and am satisfied with their adequacy and thoroughness.
---

## [Editor Report · Acceptance letter]

20 Sep 2024

PONE-D-23-36626R1 

PLOS ONE

Dear Dr. Zondag, 

I'm pleased to inform you that your manuscript has been deemed suitable for publication in PLOS ONE. Congratulations! Your manuscript is now being handed over to our production team.

Kind regards, 

on behalf of

Dr. Mubarick Nungbaso Asumah 

Academic Editor

PLOS ONE